# Electrically driven spin resonance of *4f* electrons in a single atom on a surface

Stefano Reale [1,2,3], Jiyoon Hwang[1,4], Jeongmin Oh [1,4], Harald Brune [5], Andreas J. Heinrich [1,4], Fabio Donati [1,4] ✉ & Yujeong Bae [1,4,6] ✉

A pivotal challenge in quantum technologies lies in reconciling long coherence times with efficient manipulation of the quantum states of a system. Lanthanide atoms, with their well-localized *4f* electrons, emerge as a promising solution to this dilemma if provided with a rational design for manipulation and detection. Here we construct tailored spin structures to perform electron spin resonance on a single lanthanide atom using a scanning tunneling microscope. A magnetically coupled structure made of an erbium and a titanium atom enables us to both drive the erbium's *4f* electron spins and indirectly probe them through the titanium's *3d* electrons. The erbium spin states exhibit an extended spin relaxation time and a higher driving efficiency compared to *3d* atoms with spin ½ in similarly coupled structures. Our work provides a new approach to accessing highly protected spin states, enabling their coherent control in an all-electric fashion.

The last two decades have witnessed a rising focus on the control and application of quantum coherent effects, marking the advent of the so-called "second quantum revolution". Utilizing quantum coherent functionalities of materials for novel technologies, such as imaging, information processing, and communications, requires robustness of their quantum coherence, addressability, and scalability[1]. However, these requirements often clash since decoupling the quantum states from the environment prolongs the quantum coherent properties but hinders the possibility of efficient state manipulation.

Lanthanide atoms represent a promising platform to tackle this dilemma. Their well-localized *4f* electrons show long spin relaxation $T_1$[2,3] and coherence times $T_2$[4,5]. In addition, their strong hyperfine interaction facilitates the read-out of nuclear spins[6,7]. In bulk insulators, exceedingly long $T_1$ and $T_2$ have been demonstrated using optical control and detection[8-11] down to the single atom level[12,13]. While hybrid optical-electrical approaches have been developed to access individual lanthanide atom's spins embedded in a silicon transistor[14], it is still challenging to achieve efficient control of the quantum states using electrical transport methods. This necessitates the rational design of a quantum platform capable of tackling both control and detection schemes, along with their interactions with local environments. In this context, single crystal surfaces constitute an advantageous framework both for building atomically engineered nanostructures and addressing individual spin centers, in particular using scanning probe techniques[15-18]. However, resonant driving and detection of surface-adsorbed lanthanide atoms have so far remained elusive.

In this work, we demonstrate the control and detection of *4f* electron spins by building atomic-scale structures on a surface using a scanning tunneling microscope (STM) with electron spin resonance (ESR) capabilities[19-22]. The atomic structures are composed of an erbium (Er) atom as the target spin system and a magnetically coupled titanium (Ti) atom as the sensor spin. This architecture allows us to drive ESR transitions on the Er *4f* electrons with a projected angular momentum of ℏ/2[23] and to probe them indirectly through Ti. We observed an Er $T_1$ of close to 1 μs, which is about 5 times longer than that previously measured for *3d* electrons of a remotely-driven spin-½ system on the same surface[18]. This novel platform allows for the ESR driving and read-out of the well-screened *4f* electron spin states, paving the way to integrate lanthanide atoms in quantum architectures.

[1]Center for Quantum Nanoscience (QNS), Institute for Basic Science (IBS), Seoul, Republic of Korea. [2]Ewha Womans University, Seoul, Republic of Korea. [3]Department of Energy, Politecnico di Milano, Milano, Italy. [4]Department of Physics, Ewha Womans University, Seoul, Republic of Korea. [5]Institute of Physics, Ecole Polytechnique Fédérale de Lausanne, Lausanne, Switzerland. [6]Present address: Empa, Swiss Federal Laboratories for Materials Science and Technology, nanotech@surfaces Laboratory, Dübendorf, Switzerland. ✉e-mail: donati.fabio@qns.science; bae.yujeong@qns.science

## Results

### Sensing Er Spin States through a Ti Atom

Erbium atoms on a few monolayer-thick MgO(100) on Ag(100) present a $4f^{11}$ configuration with no unpaired electrons in the $5d$ and $6s$ shells[23]. The atomic-like spin and orbital momenta are coupled through the large spin-orbit interaction into a total angular momentum $\mathbf{J}_{Er}$ with magnitude of $15\hbar/2$[23]. When adsorbed on the oxygen site of MgO (Fig. 1a), the crystal field leads to a strong hard-axis magneto-crystalline anisotropy that stabilizes a doubly-degenerate ground state with an out-of-plane component of the angular momentum $J_\perp = \pm\hbar/2$[23], which splits into two singlets when an external magnetic field (**B**) is applied. This magnetic level scheme differs from the ones of lanthanide single atom

magnets studied so far on MgO/Ag(100). For instance, dysprosium[24,25] and holmium[2,26,27] present a ground state characterized by a large $J_\perp$. The level scheme presents two lowest-lying states well separated by a significant anisotropy barrier and greatly suppresses the reversal of angular momentum, thereby stabilizing the magnetic states. Additionally, it impedes the first-order ESR transition induced by the exchange of a single quantum of angular momentum[22]. As found in a previous work[23], the component of $\mathbf{J}_{Er}$ along the magnetic field direction ($z$), defined as $J_z$, increases from $\pm\hbar/2$ to $\pm4\hbar$ by rotating **B** from the out-of-plane ($\vartheta = 0°$) to the in-plane ($\vartheta = 90°$) direction (Fig. 1b), while retaining a large probability for spin dipole transitions. Given these properties, Er can be regarded as a highly tunable two-level system allowing for

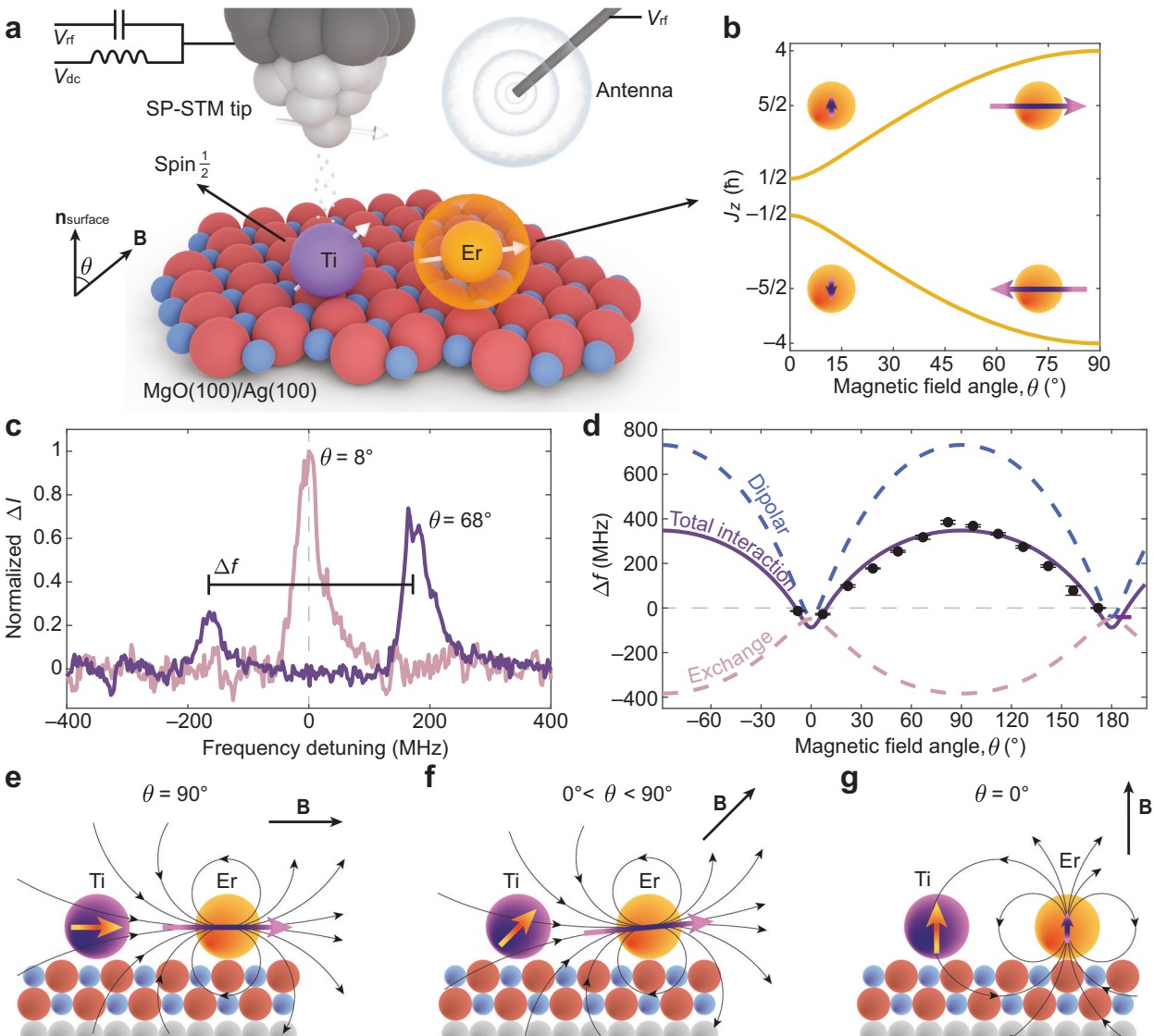

**Fig. 1 | Probing Er 4f electron spins through a Ti spin sensor. a** Schematic of the experimental set-up for ESR-STM measurement of an Er-Ti dimer built on MgO/Ag(100). The Ti atom (purple) is positioned close to the Er atom (orange) and located under a spin-polarized (SP) STM tip. The external magnetic field (**B**) defines the $z$-direction and is applied at an angle $\vartheta$ from the out-of-plane direction. **b** Projected total angular momentum of Er ($J_z$) onto the **B** field direction as a function of $\vartheta$. The strong magnetic anisotropy favors an in-plane alignment of $\mathbf{J}_{Er}$. **c** ESR spectra of the Ti atom placed 0.928 nm apart from the Er atom at different $\vartheta$. At $\vartheta = 8°$, a single ESR peak is visible (pink) while, at $\vartheta = 68°$ (purple), the two ESR peaks are separated due to the magnetic interactions between the Er and Ti. For the latter, the relative peak intensity indicates a ferromagnetic interaction (set-point: $V_{dc} = 50$ mV, $I_{dc} = 20$ pA, $V_{rf} = 12$ mV, $B = 0.3$ T). **d** ESR peak separation, $\Delta f$, as a

function of $\vartheta$. The experimental points (black dots) were acquired at different set-points ($V_{dc} = 50$ mV, $I_{dc} = 12$–30 pA, $V_{rf} = 12$–20 mV, $B = 0.3$ T). We give error bars with 95% confidence interval. The total interaction (solid purple line) calculated by the model Hamiltonian is composed of a dipolar contribution (dashed blue line) and an exchange contribution (dashed pink line). **e**–**g** Schematic of the angular momenta of Er and Ti on MgO/Ag(100). The dipolar fields induced by Er are depicted as black curved arrows. When **B** is applied along the in-plane direction ($\vartheta = 90°$), the $J_z$ is maximum and aligned with the spin of Ti giving the largest ferromagnetic dipolar interaction. When **B** is rotated, the spin of Ti follows the direction of **B** while the total angular momentum of Er is aligned preferentially in-plane (**f**). In the out-of-plane direction ($\vartheta = 0°$), $J_z$ is minimum and aligned with the spin of Ti (**g**) giving a small antiferromagnetic dipolar interaction.

efficient ESR driving. To characterize the magnetic states and aniso-tropy of Er, we utilized the dipole field sensing technique[28] with a Ti atom on the bridge binding site of MgO as a well-known spin sensor. On this binding site, Ti has a spin $\mathbf{S}_{Ti}$ of magnitude $\hbar/2$ and a relatively weak g-factor anisotropy[29] compared to the oxygen binding site[30].

We deposited Er and Ti at cryogenic temperatures (~10 K) on 2 monolayers of MgO grown on Ag(100) ("Methods" section and Fig. S1a). Their binding sites on the surface can be changed by atom manipulation (Supplementary Section 2). When isolated, a nuclear spin-free Ti atom presents a single ESR signal under an external mag-netic field (Fig. S3a). The ESR peak of Ti splits when coupled to an Er atom (Supplementary Section 4). Figure 1c shows the ESR spectra obtained on Ti in an Er-Ti dimer with the atomic separation of 0.928 nm (Fig. S2b). At the magnetic field of 0.3 T with $\vartheta = 8°$, we observed one ESR peak at the resonance frequency of Ti, which splits into two peaks separated by $\Delta f = 334 \pm 3$ MHz when rotating $\mathbf{B}$ close to the in-plane direction ($\vartheta = 68°$). The two ESR peaks stem from the magnetic interaction with the Er spin fluctuating between two states[28] during the measurement, with the relative peak intensity being pro-portional to the time-averaged population of the Er states. The pro-nounced difference in the relative intensity of the ESR peaks indicates a large imbalance in the Er state occupation even at $B = 0.3$ T and 1.3 K, which reflects the large $J_z$ of Er at $\vartheta = 68°$ (Fig. 1b). The polarity of this asymmetry depends on the character of the magnetic interactions between the two atoms (Fig. S4b, e). In Fig. 1c, the peak at the lower frequency is less intense than the one at the higher frequency and, hence, the interaction can be regarded as ferromagnetic[31], with this polarity defined as positive $\Delta f$ in Fig. 1d. We observe changes in polarity at different field directions (Fig. 1d), indicating alternating couplings between ferromagnetic and antiferromagnetic states.

The angle dependence of $\Delta f$ (Fig. 1d) gives a direct measurement of the Er-Ti interaction energy[28,31] and of the magnetic anisotropy[23]. To interpret it, we model the system through a spin-Hamiltonian includ-ing both the single atom Zeeman and anisotropy terms, as well as the interaction between the two spins:

$$H = \mu_B g_{Er} \mathbf{B} \cdot \mathbf{J}_{Er} + D J_\perp^2 + \mu_B \mathbf{B} \cdot \bar{\bar{g}}_{Ti} \cdot \mathbf{S}_{Ti} + H_{dip} + H_{exc}. \quad (1)$$

Here, $\mu_B$ is the Bohr magneton, $J_\perp$ is the out-of-plane component of $\mathbf{J}_{Er}$, $g_{Er} = 1.2$ is the Er Landé g-factor obtained from its atomic quantum numbers, and $\bar{\bar{g}}_{Ti}$ is the Ti anisotropic g-tensor[29]. We use a magnetic anisotropy parameter $D = 2.4$ meV to match the Er energy splitting found in a previous study[23]. The magnetic coupling consists of dipolar ($H_{dip}$) and Heisenberg exchange interactions ($H_{exc}$):

$$H_{dip} = \frac{\mu_0 \mu_B^2}{4\pi \hbar^2 r^3} \left[ g_{Er} \mathbf{J}_{Er} \cdot \bar{\bar{g}}_{Ti} \cdot \mathbf{S}_{Ti} - 3 \left( \hat{\mathbf{r}} \cdot g_{Er} \mathbf{J}_{Er} \right) \left( \hat{\mathbf{r}} \cdot \bar{\bar{g}}_{Ti} \cdot \mathbf{S}_{Ti} \right) \right], \quad (2)$$

$$H_{exc} = \frac{\mathscr{J}_{exc}}{\hbar^2} \left( \mathbf{J}_{Er} \cdot \mathbf{S}_{Ti} \right), \quad (3)$$

where $\mu_0$ is the vacuum permittivity, $r$ the separation between the two atoms, $\hat{\mathbf{r}}$ the unit vector connecting them[28], and $\mathscr{J}_{exc}$ the exchange interaction energy expressed in terms of $\mathbf{J}_{Er}$[32]. In our model, $\mathscr{J}_{exc}$ is the only free parameter for the fit. As shown in Fig. 1d, our model accu-rately reproduces the data for $\mathscr{J}_{exc}/h = 48$ MHz, where the positive sign indicates an antiferromagnetic coupling. This value is more than 20 times smaller than that observed for a Ti-Ti dimer at the same distance (1.16 GHz)[33]. We ascribe the smaller Er-Ti coupling to the localization of the 4f orbitals near the atom's core, which limits the overlap between Er and Ti orbitals when compared to the Ti-Ti case.

The strong angle dependence of $\Delta f$ can be understood by con-sidering the large magneto-crystalline anisotropy of $\mathbf{J}_{Er}$. At $\vartheta = 90°$, $J_z$ is at its maximum (4$\hbar$), and the angular momenta of both atoms are parallel to $\hat{\mathbf{r}}$ (Fig. 1e), which maximizes the contribution of the dipolar

coupling in a ferromagnetic configuration (positive $\Delta f$). When rotating $\mathbf{B}$ away from the in-plane direction, $\mathbf{S}_{Ti}$ follows the direction of $\mathbf{B}$, while the anisotropy of Er preserves a large component of $\mathbf{J}_{Er}$ mainly aligned along the in-plane direction (Fig. 1f). This misalignment between the two angular momenta reduces the dipolar interaction. Finally, as $\mathbf{B}$ approaches the surface normal (Fig. 1g), $\mathbf{J}_{Er}$ turns towards the out-of-plane direction with a much smaller value of $J_z = \hbar/2$. With the two momenta being perpendicular to $\hat{\mathbf{r}}$, the dipolar interaction is anti-ferromagnetic (negative $\Delta f$). Conversely, the mutual projection of $\mathbf{S}_{Ti}$ and $\mathbf{J}_{Er}$ is the only factor modulating the exchange interaction term, which remains negative $\Delta f$ (antiferromagnetic coupling with positive $\mathscr{J}_{exc}$) at all angles (dashed pink curve in Fig. 1d).

## Spin resonance of Er 4f electrons
The direct drive of ESR in STM requires positioning the tip directly on top of the target atom[19]. However, despite using a tip showing ESR signal on an isolated Ti atom (Fig. S3a), we observed no ESR when positioning the tip over an Er atom (Fig. S3b), which we attribute to the small polarization of the 5d and 6s shells of Er and to the weak inter-action between the 4f and tunneling electrons. The weak polarization of the outer shell is reflected in the absence of spin excitations in the dI/dV spectra (Fig. S1d, e, h). These factors were found to limit the tunneling magnetoresistance at the STM junction in other lanthanide atoms[24,34], possibly hindering both the ESR drive and detection[23].

To overcome this limitation, we built a strongly interacting Er-Ti dimer by positioning Ti at 0.72 nm from Er through atom manipulation (Fig. 2a and Supplementary Section 2). Similar to the isolated atom, we observed no ESR peaks at the Er position in the dimer (yellow curve in Fig. 2b). However, when the tip was positioned on Ti, we observed up to 5 peaks (pink and purple curves in Fig. 2b). The first two peaks below 10 GHz with $|\Delta f| = 2.70 \pm 0.01$ GHz correspond to the ESR transitions of Ti that were similarly found in the dimer with larger atomic separations (Fig. 1c). Hence, we label them as $f_1^{Ti}$ and $f_2^{Ti}$, respectively. In this dimer, we observed that $f_1^{Ti}$ shows a higher intensity than $f_2^{Ti}$, indicating an antiferromagnetic exchange interaction[31] (Fig. S4c, f) dominating over the dipolar coupling at this atomic separation. At higher frequencies, we further observed two peaks that are significantly blue-shifted when rotating $\mathbf{B}$ from $\vartheta = 52°$ (pink curve in Fig. 2b) to 97° (purple). The higher resonance frequencies and pronounced angle dependence indicate that those transitions involve the large and anisotropic angular momentum of Er, and, thus, we label them as $f_3^{Er}$ and $f_4^{Er}$. These transitions are not observed for all Er atoms, possibly due to the pre-sence of isotopes with large nuclear spins for which the intensity of the ESR signal is spread over multiple peaks and is below the sensitivity of our measurements (Fig. S5). In addition, their frequency separation exactly matches the one between $f_1^{Ti}$ and $f_2^{Ti}$, reflecting the same Er-Ti interaction. On the other hand, $f_3^{Er}$ and $f_4^{Er}$ are approximately equal in intensity, indicating that Ti fluctuates between two spin states with almost equal occupations. The comparable Ti states' occupation stems from the scattering with tunneling electrons and from the Zeeman splitting of Ti (~7 GHz) being smaller than the thermal energy at the measurement temperature of 1.3 K (~27 GHz). With $\mathbf{B}$ at $\vartheta = 52°$, we observed one more peak at even higher frequencies. Its frequency exactly matches the sum of $f_1^{Ti}$ and $f_4^{Er}$ (or equivalently $f_2^{Ti}$ and $f_3^{Er}$), which suggests an ESR transition involving both Ti and Er spins. We label this peak as $f_5^{TiEr}$. Remarkably, the sign of $f_3^{Er}$, $f_4^{Er}$ and $f_5^{TiEr}$ is opposite to that of $f_1^{Ti}$ and $f_2^{Ti}$, indicating a different detection mechanism for the transitions involving the Er spin, which will be discussed below. Finally, we observed an energy level crossing between Er and Ti transitions at $\vartheta \sim 12°$, with the Er resonance fre-quencies further shifting below the Ti transitions at $\vartheta \sim 0°$ (Fig. 2c and Fig. S6). This peculiar behavior is a consequence of the large difference in magnetic anisotropy between Er and Ti[23].

As shown in Fig. 2c, the angular dependence of the ESR fre-quencies is well reproduced by using Eq. 1 with $\mathscr{J}_{exc}/h = 326$ MHz for

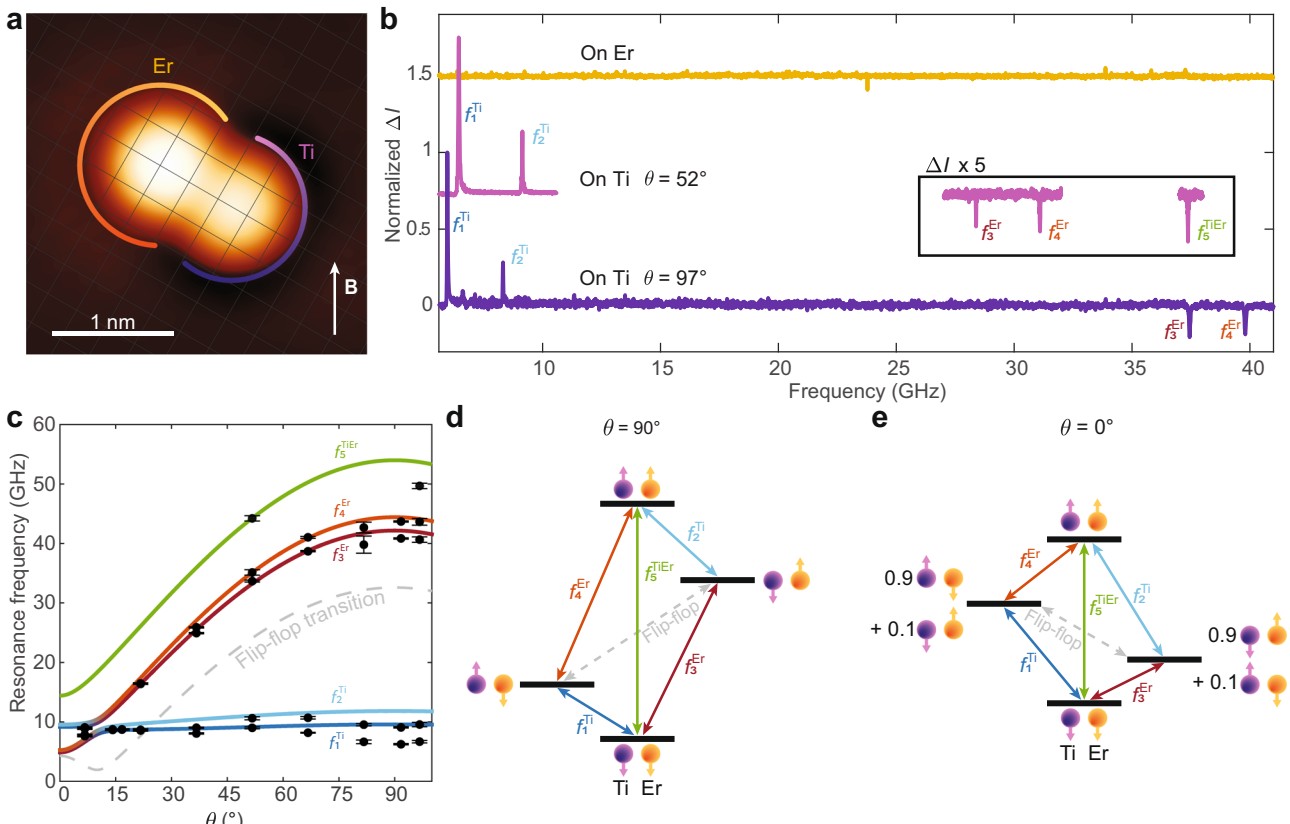

**Fig. 2 | Measurement of Er ESR transitions through a strongly coupled Ti atom.** **a** Constant-current STM image of the engineered Er-Ti dimer with the atomic separation of 0.72 nm. The intersection of grids represents the oxygen sites of MgO. The Er atom (circled in yellow) is adsorbed on the oxygen site of MgO, while the Ti atom (circled in purple) is adsorbed on the bridge site (set-point: $V_{dc} = 100$ mV, $I_{dc} = 20$ pA). **b** ESR spectra of the dimer given in **a**. When the STM tip is located on top of Er, no peaks are observed (yellow) (set-point: $V_{dc} = 50$ mV, $I_{dc} = 20$ pA, $V_{rf} = 20$ mV, $B = 0.28$ T, $\vartheta = 97°$). When the STM tip is located on top of Ti, 5 ESR peaks are detected ($f_{1,2}^{Ti}$, $f_{3,4}^{Er}$ and $f_5^{TiEr}$) with $\vartheta = 52°$ (pink), while 4 ESR peaks are detected ($f_{1,2}^{Ti}$, and $f_{3,4}^{Er}$) with $\vartheta = 97°$ (purple) (set-point: $V_{dc} = 70, 60$ mV, $I_{dc} = 30, 40$ pA, $V_{rf} = 20, 15$ mV, $B = 0.3$ T). **c** ESR frequencies as a function of $\vartheta$ at $B = 0.32$ T. The ESR frequencies obtained from each measurement are given as black dots with error bars with 95% confidence interval alongside the transition energies predicted from the model Hamiltonian for $f_1^{Ti}$ (blue line), $f_2^{Ti}$ (light blue line), $f_3^{Er}$ (red line), $f_4^{Er}$ (orange line), $f_5^{TiEr}$ (green line), and flip-flop transition (dashed gray line). The experimental points were obtained at different set-points ($V_{dc} = 60—70$ mV, $I_{dc} = 12—40$ pA, $V_{rf} = 15—25$ mV, $B = 0.28—0.8$ T); the resonance frequencies were rescaled by $0.32$ T/$B$. **d, e** Four-level schemes corresponding to the energies of the 4 spin states of the Er-Ti dimer and the corresponding transitions depicted as colored arrows at $B = 0.32$ T with different $\vartheta$ (90° and 0°, respectively). At $\vartheta = 90°$ (**d**), the spin states are given by the Zeeman product states, while at $\vartheta = 0°$ (**e**), a linear combination of the Zeeman product states is needed to describe the levels.

this Ti-Er pair with 0.72 nm separation. We observed small deviations for $f_1^{Ti}$, $f_2^{Ti}$ and $f_5^{TiEr}$, which we ascribe to different experimental conditions and magnetic interaction of Ti with the tip, which is not included in our model. Diagonalizing the Hamiltonian in Eq. 1 allows us to analyze the quantum states of the Er-Ti dimer in terms of individual Er and Ti spin states. For an in-plane $B = 0.3$ T, the energy detuning between the Er and Ti spins (30 GHz) is much larger than the interaction energy (about 3 GHz). Therefore, the Er-Ti dimer can be modeled with the 4 Zeeman product states of the Er and Ti spins. Following this picture, we can support the assignment of $f_{1,2}^{Ti}$ as Ti spin transitions occurring with no changes in the Er state, while $f_{3,4}^{Er}$ correspond to Er spin transitions without altering Ti. Finally, we attribute $f_5^{TiEr}$ to a double-flip transition involving both Er and Ti spins. Even though a $|\triangle m| > 1\hbar$ process is generally forbidden to first order, anisotropic terms in the magnetic interaction can give rise to higher order matrix elements connecting states with $\Delta m = \pm 2\hbar$[35].

When the field is oriented at $\vartheta = 0°$, both $\mathbf{J}_{Er}$ and $\mathbf{S}_{Ti}$ show an expectation value of $\hbar/2$, but a detuning still occurs due to the difference between the out-of-plane g-factors, $g_{Er} = 1.2$ and $g_{Ti} = 1.989 \pm 0.024$[29]. This detuning is comparable to their interaction energy and, thus, the two middle levels are no longer described by Zeeman product states (Fig. 2e). Finally, at the level crossing angle ($\vartheta \sim 12°$), the two Er and Ti middle levels become singlet and triplet

states[33]. However, measuring ESR spectra under these conditions becomes challenging (Fig. S7), possibly due to the limitation in our detection as discussed in the following.

## Erbium ESR detection and driving mechanisms

The detection of ESR peaks exclusively occurs when the tip is positioned on top of Ti. Moving the tip from Ti to Er, the intensities of $f_3^{Er}$ and $f_4^{Er}$ gradually decrease and eventually vanish at ~0.3 nm from the Ti center (Fig. S8). This behavior indicates that driving an ESR transition on Er must induce a change in the Ti state occupation, subsequently modifying the spin polarization of the tunnel junction. In addition, while the Ti transitions always yield positive peaks $f_{1,2}^{Ti}$, Er ESR signals differ depending on specific tip conditions, i.e., different tips show positive or negative sign for $f_{3,4}^{Er}$ (Fig. 3a).

To further delve into the driving and detection mechanisms of the Er spin, we measured the intensities of $f_1^{Ti}$ and $f_3^{Er}$ as a function of $V_{rf}$ using a tip that shows negative Er peaks (Fig. 3b). While $f_1^{Ti}$ exhibits a continuous increase in intensity with increasing $V_{rf}$, $f_3^{Er}$ reaches saturation at $V_{rf} \sim 20$ mV. The result for $f_1^{Ti}$ aligns with previous measurements on Ti[33], while the low-power saturation of Er is comparable to that of Fe, which might reflect a long $T_1$ and/or a high Rabi rate ($\Omega$)[36]. To understand this $V_{rf}$-dependence as well as the signs of ESR signals, we developed a rate equation model (Supplementary Section 7) based

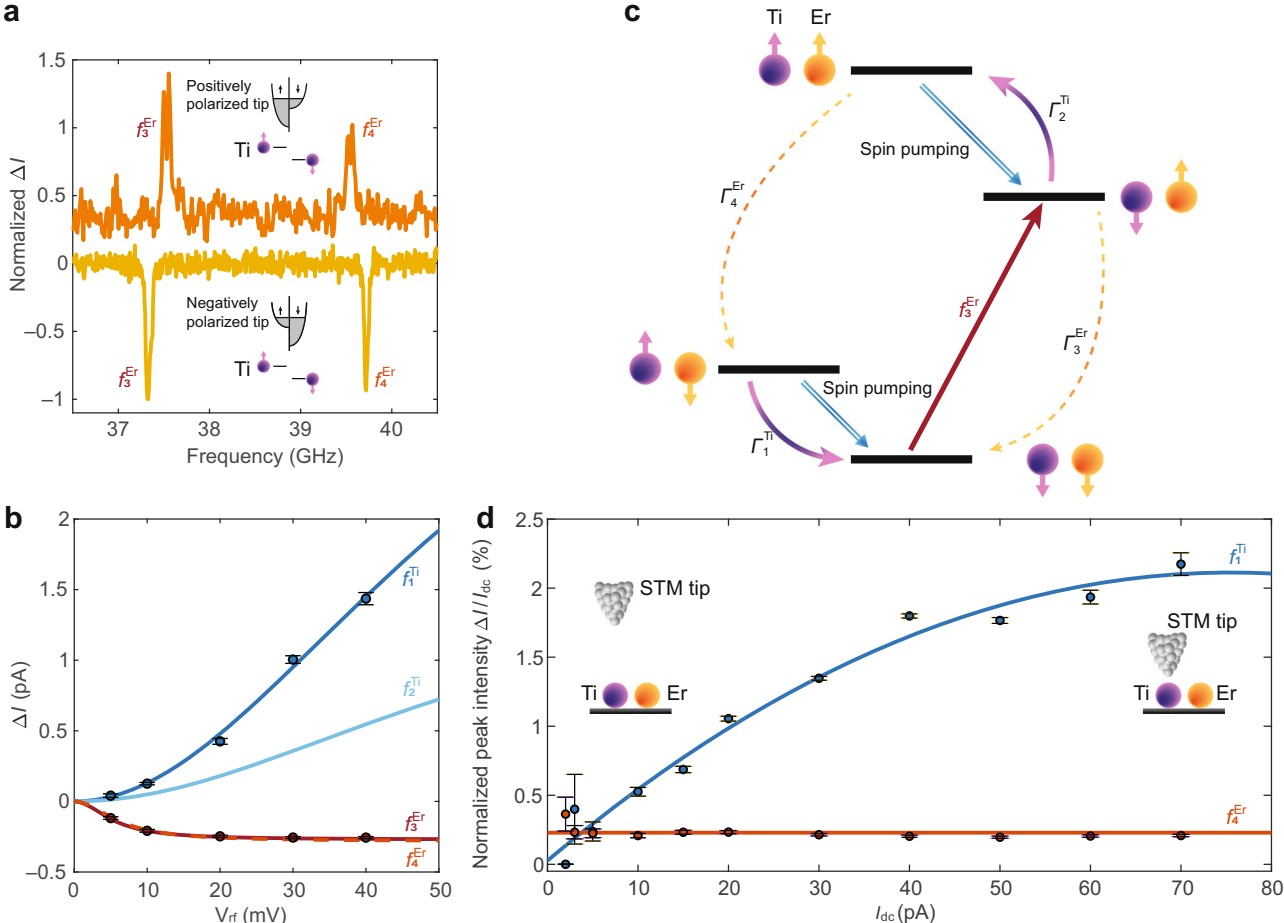

**Fig. 3 | Detection and driving mechanisms of Er ESR transitions. a** ESR spectra showing $f_{3,4}^{Er}$ for two different STM tips: negative peaks related to negative spin pumping (yellow line) and positive peaks related to positive spin pumping (orange line) (set-point: $I_{dc} = 12$, 20 pA, $V_{dc} = 70$ mV, $V_{rf} = 25$ mV, $B = 0.28$, 0.32 T, $\vartheta = 67°$). **b** ESR peak intensities as a function of $V_{rf}$. The measured values for $f_1^{Ti}$ and $f_3^{Er}$ are given by black dots while the intensities predicted from the rate equation model (Supplementary Section 9) for $f_{1,2}^{Ti}$ and $f_{3,4}^{Er}$ are given as blue, light blue, red solid lines and an orange dashed line, respectively (set-point: $I_{dc} = 40$ pA, $V_{dc} = 70$ mV, $B = 0.28$ T, $\vartheta = 97°$). **c** Four-level scheme explaining the rate equation model while

driving $f_3^{Er}$ (red arrow). The Ti's spin relaxation rates $\Gamma_1^{Ti}$ and $\Gamma_2^{Ti}$ are depicted as purple arrows while the Er spin relaxation rates $\Gamma_3^{Er}$ and $\Gamma_4^{Er}$ are given as dashed yellow arrows. The negative spin pumping effect is represented as blue double arrows. **d** Normalized ESR peak intensities ($\Delta I/I_{dc}$) for $f_1^{Ti}$ (blue circles) and for $f_4^{Er}$ (orange circles) at different tip heights. Here, the tip height is controlled by the set-point current $I_{dc}$ (set-point: $V_{dc} = 70$ mV, $V_{rf} = 10$ mV, $B = 0.28$ T, $\vartheta = 97°$). The blue and orange lines serve as guides for the eye. The insets show two different tip-Ti distances: larger for lower $I_{dc}$ and smaller for higher $I_{dc}$. In **b** and **d**, the error bars are given with 95% confidence interval.

on the four-level scheme depicted in Fig. 3c. When driving $f_3^{Er}$ (red arrow), the populations of the initial and final states involved in the transition tend to equalize through a population transfer[37]. The changes in population are counteracted by the relaxation rates of each state ($\Gamma_{1,2}^{Ti}$ and $\Gamma_{3,4}^{Er}$), which tend to repopulate the depleted states. These rates are inversely proportional to the $T_1$ of the atom involved in the spin flip. These relaxations happen due to an exchange of energy with the environment which tends to relax the populations towards the thermal equilibrium. An exchange of energy to (from) the environment leads to a transition to a lower (higher) energy level. Since Ti located under the tip is strongly influenced by tunneling electrons, relaxation events occur on a much shorter timescale than for Er[38], providing a more efficient pathway to attain the steady state. In addition, to account for the tip-dependent sign and intensity of Er ESR signals, we included a spin pumping term originating from the spin-polarized tunnel current (Fig. 3c for a negatively polarized tip)[17,39]. In inelastic scattering events, the exchange of angular momenta occurs while retaining the total angular momentum of the system[39]. That is, the spin-polarized tunneling electrons lead to scattering events with preferential polarization, as depicted in the inset of Fig. 3a, c. Thus, the tunneling electrons can shift the Ti spin occupation altering the population balance with

respect to the thermal equilibrium (see Supplementary Section 9). The proposed detection scheme based on the change of Ti state population accurately describes the $V_{rf}$-dependence (Fig. 3b) and the tip-dependent sign variations of the ESR signals (Fig. S10b).

Finally, to identify the ESR driving source of the Er spin, we follow the relative peak intensity ($\Delta I/I_{dc}$) at different tip heights, as controlled by $I_{dc}$. As shown in Fig. 3d, $\Delta I/I_{dc}$ of $f_1^{Ti}$ increases with reducing the tip-sample distance (increasing $I_{dc}$), indicating that the main driving term for Ti arises from the exchange interaction with the spin-polarized tip[40,41]. On the other hand, $\Delta I/I_{dc}$ for $f_4^{Er}$ remains independent of $I_{dc}$, which identifies the modulation of the magnetic interaction with Ti as the ESR driving source of Er[42]. The modulation of the magnetic coupling[43], in combination with anisotropic interaction terms[35], additionally explains the drive of the double-flip transition $f_5^{TiEr}$.

### Relaxation time measurement through electron-electron double resonance

As previously discussed about the Er-Ti dimer with 0.928 nm separation, the relative peak intensity of the Ti peaks $f_1^{Ti}$ and $f_2^{Ti}$ reflects the time averaged population of the Er states (Fig. 4a), i.e., changes in the time-averaged spin states of Er will induce the change of the relative peak

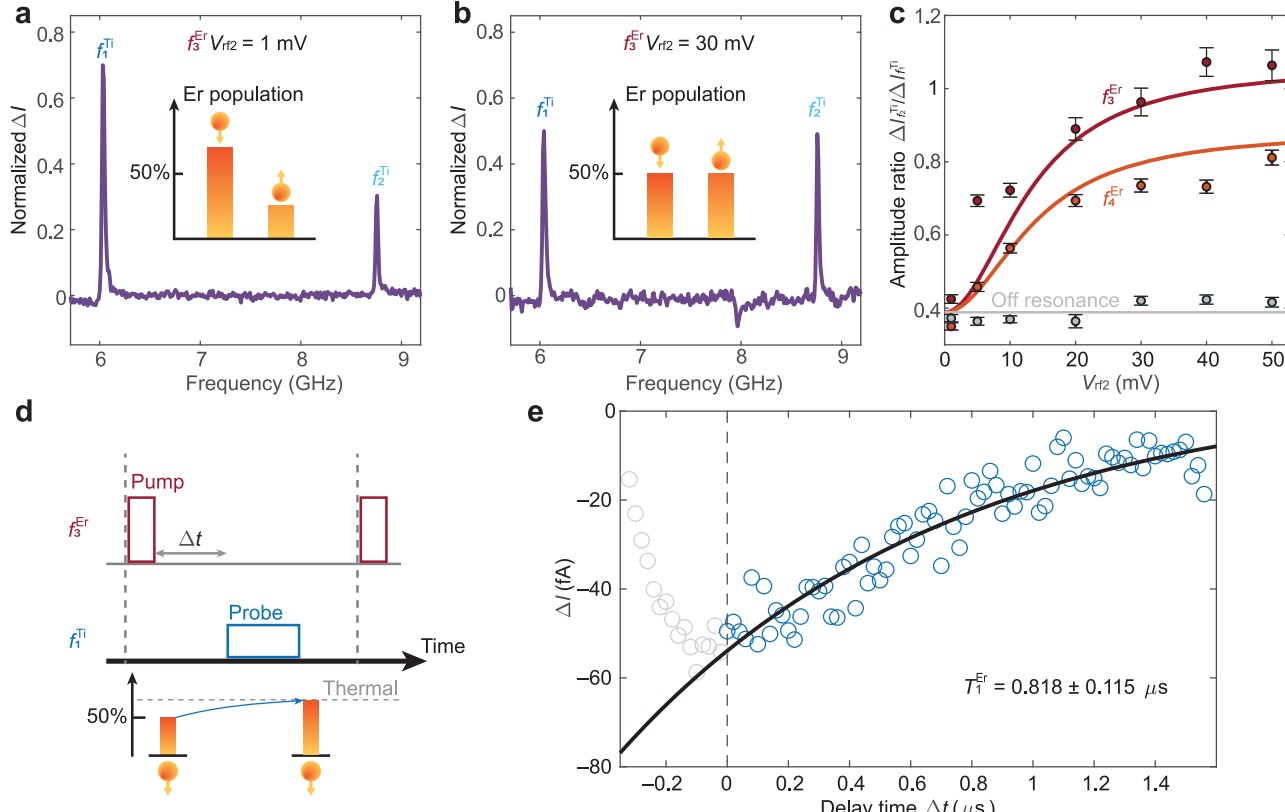

**Fig. 4 | Determination of Er spin relaxation time. a, b** Double resonance spectra in the frequency range covering Ti ESR transitions $f_{1,2}^{Ti}$ (**a**) without and (**b**) with simultaneous driving of Er at the ESR frequency of $f_3^{Er}$. The peak intensities of $f_{1,2}^{Ti}$ are related to the relative population of the Er spin states (insets). The spectra were normalized to the sum of their peak intensity. **c** ESR intensity ratios between $\triangle I_{f_1}^{Ti}$ and $\triangle I_{f_2}^{Ti}$ as a function of the driving strength $V_{rf2}$ at different Er ESR transition states (red, orange, and gray circles for $f_3^{Er}, f_4^{Er}$, and off-resonance, respectively). The solid curves show the correspondent simulation results by the rate equation model (Supplementary Section 9) for $f_3^{Er}$ (red line), $f_4^{Er}$ (orange line) and at an off-resonance frequency (gray line). The experimental points are given as black dots with error bars corresponding to 95% confidence interval. Set-point: $I_{dc} = 15$ pA,

$V_{dc} = 70$ mV, $V_{rf} = 30$ mV, $V_{rf2} = 1$–50 mV, $B = 0.28$ T, $\vartheta = 97°$. **d** Schematics of the inversion recovery measurement in a pump-probe pulse scheme to determine the Er spin relaxation time $T_1^{Er}$. Each sequence is composed of a pump pulse at the resonance frequency of $f_3^{Er}$ (red box) and a probe pulse at the resonance frequency of $f_1^{Ti}$ (blue box). The probe pulse follows the pump pulse after a delay time $\Delta t$. The population of the Er states after the pump pulse relaxes back to the thermal state following its $T_1$. **e** The experimental data for the inversion recovery measurement (blue circles) show the intensity of the ESR signal at the probe pulse $f_1^{Ti}$ as a function of $\Delta t$. The black line shows the fit using an exponential function with $T_1^{Er}$ of about 1 μs. Set-point: $I_{dc} = 50$ pA, $V_{dc} = 70$ mV, $V_{rf\ pump} = 60$ mV, $V_{rf\ probe} = 100$ mV, $B = 0.28$ T, $\vartheta = 97°$.

intensity of $f_1^{Ti}$ and $f_2^{Ti}$. Thus, by monitoring the Ti ESR signals, we can observe the evolution of the Er spin state. In this way, we characterize the characteristic relaxation time $T_1^{Er}$ by exciting the Er into an out-of-equilibrium state and monitoring its relaxation towards the thermal state.

By applying an additional rf voltage ($V_{rf2}$), Ti and Er spins can be simultaneously driven in the so-called "electron-electron double resonance" scheme[44]. With this scheme, it is possible to drive Er transitions and sense the change in population through the Ti ones. For instance, in double resonance experiment, the relative intensities of $f_1^{Ti}$ and $f_2^{Ti}$ are equalized when $f_3^{Er}$ is simultaneously driven (Fig. 4b). As shown in Fig. 4c, the intensity ratio of $f_1^{Ti}$ and $f_2^{Ti}$ ($\Delta I_{f_2}^{Ti}/\Delta I_{f_1}^{Ti}$) increases with increasing $V_{rf}$ only when $V_{rf2}$ is applied at the resonance frequency of $f_3^{Er}$ or $f_4^{Er}$, enabling selective modulation of the Er states to an out-of-equilibrium configuration.

Taking advantage of this selective driving mechanism, we implemented an inversion recovery measurement to estimate the spin relaxation time of Er ($T_1^{Er}$) in a pump-probe scheme (Fig. 4d). After exciting $f_3^{Er}$ with a pumping rf pulse of 200 ns duration that equalized the Er population, we applied a probe pulse of 500 ns for $f_1^{Ti}$ after a delay time $\Delta t$. Using this sequence, we monitored the time evolution of the intensity of $f_1^{Ti}$ as a function of $\Delta t$ from the out-of-equilibrium to the thermal state (Fig. 4e). The fit to an exponential function (Fig. 4e) gives $T_1^{Er} = 0.818 \pm 0.115$ μs, which is five times longer than that

previously measured for Fe-Ti dimers in the absence of tunnel current[18]. We attribute this enhancement to the efficient decoupling of $4f$ electrons from the environment, which reduces the relaxation events arising from the scattering with substrate electrons.

The $T_1^{Er}$ observed through Ti likely differs from the intrinsic relaxation time of Er on this surface due to its strong interaction with the Ti atom. Nevertheless, the large $T_1^{Er}$ indicates that the rapid spin fluctuations of Ti occurring on the timescale of a few ns[38] do not significantly perturb the stability of the Er states. This property partially originates from the large energy detuning between Er and Ti levels, which prevents the energy exchange required for spin-flip events. Using the experimentally obtained value of $T_1^{Er}$ in the rate equation model, we extract a driving term $W = \Omega^2 T_2/2$ for Er that is two times larger than for Ti in the same dimer (Supplementary Section 9). Despite the long spin lifetime and large driving term, attempts to drive Er Rabi oscillations through Ti do not yield a complete cycle (Fig. S11b), preventing a direct measure of the Er $T_2$. This is most likely due to a relatively low Rabi rate $\Omega$ provided by the moderate Er-Ti exchange coupling, which is about 2—3 times smaller than in the Fe-Ti dimer[42]. In turn, a low value of $\Omega$ together with a large driving term $W$ would imply much longer $T_2$ for Er than previous $3d$ elements, highlighting the potential of $4f$ electrons to realize higher performance atomic-scale qubits.

## Discussion

We demonstrated a new experimental approach to electrically drive ESR on the elusive $4f$ electrons in a surface-adsorbed lanthanide atom with long spin relaxation time. Given the reduced scattering with the substrate electrons, it is reasonable to anticipate an enhancement in the coherence time of Er in comparison to $3d$ elements. This allows one to develop more advanced pulse sequences for quantum coherent manipulation on atomic-scale spin platforms. We expect that, by employing a similar approach to different atomic structures, we can reduce the influence of the spin fluctuations of the atom used for the detection and amplify the ESR driving on the $4f$ electrons, enabling the use of lanthanide atoms as surface spin qubits with superior properties compared to the routinely adopted $3d$ elements.

## Methods

### STM measurements

Our experiment was performed in a home-built STM operating at the cryogenic temperature of ~1.3 K in an ultrahigh vacuum environment ($<1 \times 10^{-9}$ Torr)[45]. Using a two-axis vector magnet (6 T in-plane/4 T out-of-plane), the magnetic fields were varied from 0.28 T to 0.9 T at different angles from the surface normal[45]. To allow atom deposition on the sample kept in the STM stage, the sample is slightly tilted from the axis of the magnet by ~7° as estimated from the fit to the data shown in Fig. 1d. Considering this misalignment, all our experimental $\vartheta$ were offset by that amount accordingly. The magnetic tips used in our measurements were prepared by picking up ~4—9 Fe atoms from the MgO surface until the tips presented good ESR signals on isolated Ti atoms.

### ESR measurements

We used two different schemes to apply $V_{rf}$ to the STM junction: one through the tip and one through an antenna (rf generators: Keysight E8257D and E8267D)[45]. In all our measurement involving a single rf sweep, we applied the $V_{rf}$ using an antenna located near the sample[45] except for the data in Fig. 3b, where the $V_{rf}$ was combined with the dc bias voltage $V_{dc}$ using a bias tee at room temperature and then applied to the STM tip. The data in Fig. 4a—c were acquired by applying $V_{rf}$ to the tip and simultaneously $V_{rf2}$ to the antenna. For the measurements reported in Figs. 4e and S11, the two rf voltages ($V_{rf}$ and $V_{rf2}$) were combined through a power splitter (minicircuits ZC2PD-K0244 + ) and applied to the STM tip. For these measurements, both rf generators were gated by an arbitrary waveform generator (Tektronix, AWG 70002B).

### Sample preparation

The surface of a Ag(100) substrate was cleaned by repeated cycles of Ar+ sputtering and annealing (700 K). We grew atomically thin layers of MgO(100) on the Ag(100) by evaporating Mg metal in an oxygen atmosphere with partial pressure of $10^{-6}$ Torr while maintaining the sample at a temperature of ~590 K, following a procedure described in a previous work[46]. We deposited Fe, Ti and Er atoms (<1% of monolayer) from high purity rods (>99%) using an e-beam evaporator. During the deposition the sample was held at ~10 K in order to have well-isolated single atoms on the surface. As described in previous works, the Ti atoms on 2 ML MgO/Ag(100) show a spin magnitude of $\hbar/2$ with g-factor anisotropy[29], a behavior previously attributed to hydrogenation[31].

### Analysis of ESR spectra

We fit the ESR spectra using a model given in[33] in order to extract the resonance frequency, peak intensity, and peak width for the data shown in Figs. 1d, 2c, 3b, d, and 4c.

### ESR spectra normalization

Figure 1c: The spectrum at $\vartheta = 8°$ (pink) was normalized at its maximum intensity, while the spectrum at $\vartheta = 68°$ (purple) was normalized to the sum of the intensities of its two peaks. The frequency detuning is defined with respect to 9.1 GHz (8.1 GHz) for the spectrum at $\vartheta = 8°$ ($\vartheta = 68°$).

Figure 2b: The spectra measured on Ti at $\vartheta = 52°$ (pink) and at $\vartheta = 97°$ (purple) were normalized at their respective maxima, while the spectrum measured on top of Er was rescaled by the same amount used for the spectrum measured on Ti at $\vartheta = 97°$. The spectra measured on Ti at $\vartheta = 52°$ and on Er are offset for clarity.

## Reporting summary

Further information on research design is available in the Nature Portfolio Reporting Summary linked to this article.

## Data availability

The data used in this study are available in the figshare database under accession code [https://doi.org/10.6084/m9.figshare.24190884].

## Code availability

The code used to plot Figs. 1d, 2c, 3b, and 4c are available alongside the relative data at https://doi.org/10.24433/CO.9869055.v1.

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

## Acknowledgements

We thank Taehong Ahn and Leonard Edens for their support at the initial stage of the experiment and Yi Chen, Arzhang Ardavan, and Joaquín Fernández-Rossier for fruitful discussions. We acknowledge support from the Institute for Basic Science (IBS-R027-D1). Y.B. acknowledges support from Asian Office of Aerospace Research and Development (FA2386-20-1-4052). H.B. acknowledges funding from the SNSF AdG (TMAG-2_209266).

## Author contributions

S.R. and F.D. conceived the idea. S.R., F.D., and Y.B. designed the experiment. S.R., J.H., J.O., and Y.B. performed the experiments. S.R. analyzed the data and developed the model with the supervision of F.D. and Y.B.; H.B., A.J.H., F.D., and Y.B. supervised the project. S.R., F.D., and Y.B. wrote the manuscript with contributions from all authors. All authors discussed the results.

## Competing interests

The authors declare no competing interests.
