## [Peer Review File · Nature Communications]

Reviewers' Comments:

Reviewer #2:

Remarks to the Author:

In the manuscript titled "Electrically Driven Spin Resonance of 4f Electrons in a Single Atom on a Surface" submitted by Reale et al., the authors perform ESR-STM experiments on hydrogenated Ti to drive spin resonance in a nearby 4f atom, namely Erbium. Experiments were performed at different inter-atomic distances of this dimer and it was tried to learn about the T1 and T2 times as well as the driving force of the 4f electrons via this remote ESR-STM technique.

Overall the manuscript is very well and clearly written and the figures are clear. The performed experiments are sound and explained well. Comparing the work to other ESR-STM related publications in this journal, I would adjudicate the presented manuscript of similar quality, importance and novelty. Hence, I believe Nature Communications is a good fit for this manuscript.

However, I do have some points that have to be addressed by the authors prior considering the manuscript for publication. Please read below:

1. Line 40: Reference 18 is a wrong link. But should be updated to be pointing towards the recent Science publication of the group (<https://www.science.org/stoken/author-tokens/ST-1487/full>)
2. While the introduction focuses a lot on the strategy to improve coherence times and driving mechanisms of spin states and use 4f states for this purpose, the later results in the manuscript cannot hold up to this anymore. In the end, the coherence time could not directly be measured (probably due to the weak coupling) and the driving force was only estimated, as the authors state. Therefore, I suggest to soften the claims given in the introduction, especially towards the end, a bit. Or to shift it more towards the successful experiments that are presented. Another, maybe additional, option could also be to bring parts of the supplement into the main text and show, e.g. Fig S8 in the main paper. This way more of the discussion about this topic enters the main text and is not omitted so quickly towards the end. I also would appreciate more clear statements about what can be said or not from the presented experiments in the conclusions. If more experiments need to be done in the future this is exciting and something to be written.
3. I am wondering whether in the presented manuscript the same Ti species as previously and also in the recent Science publication (ref. 18) has been used. In the latter, the authors mention that the Ti is a hydrogenated atom, whereas this is not mentioned anymore in the present manuscript. Is there new insights that the Ti is not hydrogenated, or was it just not mentioned? This should be clarified.
4. For example in Forrester et al., a 4f atom (Ho) was directly addressed via SP-STM methods. I assume the authors did, but don't mention it: Did the authors try different elements, e.g. also Ho, to drive the ESR directly and not remotely via a Ti spin? If so, was it not successful? Besides the remote driving, I believe this would be very exciting, despite the probable reduction in T1/T2 times. A brief statement about this in the manuscript would be very helpful for the whole community.
5. The authors write about a 5x longer T1 time compared to 3d atoms measured on the same surface. However, while I agree with the general statement, I find it a bit too much to directly put numbers to these measured times. There are many parameters at play that can be different in these two scenarios that I would not say this can be a 1:1 comparison. Magnetic moments, couplings, the actual applied magnetic field, the state mixing, influences of the tip to name a few. I would ask for a more general statement of this finding.
6. Maybe I am misunderstanding, but in Figure 1c in the case of AFM coupling, should the first peak not only be smaller in the regime where J is smaller than B? Here we have a coupling of 400

MHz but a magnetic field of 0.3T, which at $1\mu_B$ already corresponds to about 4.6GHz of Zeeman splitting. Could the authors please clarify my confusion here?

7. In Figure 3c: How do I have to picture an upward relaxation mechanism (Γ_2)? Or are the levels drawn there no energy levels?

8. Line 167: What are the significant digits and error bars of both of these g-factors? How were they determined? And as for the latter one, wasn't it reported to be anisotropic?

9. When reading lines 149/150 and 200, it is not fully clear to me how the peaks/dips behave for the ESR signal of the Er. Are the Ti and Er peaks always opposite (peak vs dip)? Or is the Er signal tip dependent and can be both, either a peak or a dip?

10. Line 219: Should the f_{Ti} not be f_{Er} ?

Reviewer #3:

Remarks to the Author:

The authors present in this work an experimental approach to drive electron-spin resonance (ESR) on Er atoms (lanthanide atoms with 4f electrons) and find relatively long spin relaxation times, as compared with atoms of transition elements. The approach is based on the coupling of Er atoms with Ti atoms, which act as detectors.

The main conclusion of this work is of great importance for the ESR community and widens the possibilities to use single atoms to do electrically driven quantum coherent control of single spins on surfaces. The work is scientifically sound, the manuscript is well written, and the main conclusions are clearly stated. So, in this sense, I am certainly inclined to recommend the publication of this manuscript provided several issues are clarified or better described in a revised version. Below, I proceed to list those issues.

1) For didactic reasons, and as a reference, it would have been good to add in the main manuscript ESR spectra of isolated Ti atoms. The authors could at least mention in a revised version that those spectra are reported in Fig. S3 in the SI. In general, I think that the authors could do a better job in the manuscript specifying where to find the information in the SI.

2) As the authors explained in the manuscript, no direct ESR signal was possible to acquire when they placed the SP-STM tip on top of the Er atoms. They also speculate about the reason for that. Maybe they can elaborate a bit more using the information provided in Fig. S1, namely it would be interesting to know if the differential conductance spectra in the absence of microwaves already hint at the absence of an ESR signal when the microwaves are applied.

3) From the text in the manuscript, it is a little bit unclear what the authors mean by a term originating from the spin-polarized current in the context of the rate equation model summarized in Fig. 3c. I know that they elaborate on this in section 7 of the SI, but readers would benefit from a more detailed/didactic explanation of the meaning of this spin pumping.

4) Related to the previous issue, in the discussion of the rate equation model (section 7 of the SI), it is a bit unclear to me how the solution of the rate equation is finally converted into the ESR signal (ΔI in Fig. S7). Could the authors elaborate more in the SI about this issue?

5) The authors measure the relaxation time in the Er atoms with a sophisticated protocol that I must admit I have not quite understood. Could the authors make this discussion at the beginning of section "Relaxation Time Measurement ..." somehow more accessible to non-specialists?

6) Related to the previous issue, the authors report the Er relaxation time as a single number. Should I understand that this is the intrinsic relaxation time of Er on this surface? How can one avoid the influence of the Ti atom?

Reply to Reviewers

Reviewer #2 (Remarks to the Author):

In the manuscript titled “Electrically Driven Spin Resonance of 4f Electrons in a Single Atom on a Surface” submitted by Reale et al., the authors perform ESR-STM experiments on hydrogenated Ti to drive spin resonance in a nearby 4f atom, namely Erbium. Experiments were performed at different inter-atomic distances of this dimer and it was tried to learn about the T1 and T2 times as well as the driving force of the 4f electrons via this remote ESR-STM technique.

Overall, the manuscript is very well and clearly written, and the figures are clear. The performed experiments are sound and explained well. Comparing the work to other ESR-STM related publications in this journal, I would adjudicate the presented manuscript of similar quality, importance, and novelty. Hence, I believe Nature Communications is a good fit for this manuscript.

However, I do have some points that have to be addressed by the authors prior considering the manuscript for publication. Please read below:

1. Line 40: Reference 18 is a wrong link. But should be updated to be pointing towards the recent Science publication of the group (<https://www.science.org/stoken/author-tokens/ST-1487/full>)

Reply: We appreciate the reviewer’s careful reading and valuable comments with her/his support for our paper. We corrected the reference and updated the link.

2. While the introduction focuses a lot on the strategy to improve coherence times and driving mechanisms of spin states and use 4f states for this purpose, the later results in the manuscript cannot hold up to this anymore. In the end, the coherence time could not directly be measured (probably due to the weak coupling) and the driving force was only estimated, as the authors state. Therefore, I suggest to soften the claims given in the introduction, especially towards the end, a bit. Or to shift it more towards the successful experiments that are presented. Another, maybe additional, option could also be to bring parts of the supplement into the main text and show, e.g. Fig S8 in the main paper. This way more of the discussion about this topic enters the main text and is not omitted so quickly towards the end. I also would appreciate more clear statements about what can be said or not from the presented experiments in the conclusions. If more experiments need to be done in the future this is exciting and something to be written.

Reply: We agree with the reviewer. We corrected the manuscript by softening the claims. The corresponding part in abstract reads "The erbium spin states exhibit extended spin relaxation time and a higher driving efficiency compared to the 3d atoms with spin ½ in similarly coupled structures." In the introduction, we changed "coherent manipulation" to "resonant driving" (page 2).

We appreciate the reviewer's suggestion about moving contents (e.g. Fig. S8) from supplementary materials. However, the model descriptions in the SI might be too specific for a broad readership. In addition, while Ti shows clear Rabi oscillations in Fig. S8, we could not observe such oscillations for Er. Since it is unclear whether this absence of Rabi oscillations is due to the initialization of Er spins, to insufficient driving of Er, or to something else, we decided to keep this figure as a part of the supplementary information.

In addition, we added a short perspective to the conclusions at page 13: "This allows one to develop more advanced pulse sequences for quantum coherent manipulation on atomic-scale spin platforms. We expect that, by employing a similar approach in different atomic structures, we can reduce the influence of the

spin fluctuations of the atom used for the detection and amplify the ESR driving on the 4f electrons, enabling the use of lanthanide atoms as surface spin qubits with superior properties compared to the routinely adopted 3d elements."

3. I am wondering whether in the presented manuscript the same Ti species as previously and also in the recent Science publication (ref. 18) has been used. In the latter, the authors mention that the Ti is a hydrogenated atom, whereas this is not mentioned anymore in the present manuscript. Is there new insights that the Ti is not hydrogenated, or was it just not mentioned? This should be clarified.

Reply: The reviewer is correct. The Ti species is the same as in previous ESR-STM works. While the hydrogenation of Ti has been supported by previous works to explain the spin-1/2 state of Ti on MgO, recent studies suggest other possibilities, such as charge transfer. Since this point is still under debate, we originally preferred not to specify the hydrogenation of Ti atoms. However, we agree with the reviewer on the need of being consistent with other recent works. Hence, we added one sentence in Methods at page 14, which reads: "As described in previous works the Ti atoms on 2 ML MgO/Ag(100) shows spin 1/2 behavior which is presumably originating from hydrogenation."

4. For example in Forrester et al., a 4f atom (Ho) was directly addressed via SP-STM methods. I assume the authors did, but don't mention it: Did the authors try different elements, e.g. also Ho, to drive the ESR directly and not remotely via a Ti spin? If so, was it not successful? Besides the remote driving, I believe this would be very exciting, despite the probable reduction in T1/T2 times. A brief statement about this in the manuscript would be very helpful for the whole community.

Reply: We appreciate the reviewer's suggestion. While we focus on the potential use of 4f atoms on surfaces as a qubit, Ho on MgO is known as a single atom magnet. We haven't tried to do ESR driving on Ho, but we expect the resonant driving of the Ho to be quite challenging or even not feasible since the ground state of Ho has a large magnetic quantum number and the two lowest-lying states are separated by a large anisotropy barrier. We added a brief statement about this issue in the revised manuscript (page 2), which reads: "This magnetic level scheme differs from the ones of lanthanide single atom magnets studied so far on MgO/Ag(100). For instance, dysprosium and holmium present a ground state characterized by a large J_{\square} . The level scheme presents two lowest-lying states well separated by a significant anisotropy barrier and greatly suppresses the reversal of angular momentum, thereby stabilizing the magnetic states. Additionally, it impedes the first-order ESR transition induced by the exchange of a single quantum of angular momentum." In addition, we added the reference mentioned by the reviewer as the new reference 27.

5. The authors write about a 5x longer T1 time compared to 3d atoms measured on the same surface. However, while I agree with the general statement, I find it a bit too much to directly put numbers to these measured times. There are many parameters at play that can be different in these two scenarios that I would not say this can be a 1:1 comparison. Magnetic moments, couplings, the actual applied magnetic field, the state mixing, influences of the tip to name a few. I would ask for a more general statement of this finding.

Reply: The reviewer is totally correct. We revised the statement in the abstract, which now reads: "The erbium spin states exhibit an extended spin relaxation time and a higher driving efficiency compared to 3d

atoms with spin $\frac{1}{2}$ in similarly coupled structures." In addition, in the introduction (page 2), we specified the case of comparison as "a remotely-driven spin- $\frac{1}{2}$ system" in the sentence starting "We observed an Er T_1 of close to $1 \mu\text{s}$, ..."

6. Maybe I am misunderstanding, but in Figure 1c in the case of AFM coupling, should the first peak not only be smaller in the regime where J is smaller than B ? Here we have a coupling of 400 MHz but a magnetic field of 0.3T, which at $1\mu\text{B}$ already corresponds to about 4.6GHz of Zeeman splitting. Could the authors please clarify my confusion here?

Reply: To better explain the effect of the Er-Ti interaction on the Ti ESR peak intensity, we added a new section (now Supplementary Section 4) in the SI, entitled "Effect of the Er-Ti coupling on the ESR spectra of Ti". In this section, we introduce the total interaction tensor \bar{J}_{int} (Eq. S2) as the sum of the dipolar tensor (Eq. S1) and exchange interaction energy (Eq. 2). For a fixed Er-Ti separation at a constant magnetic field angle, this interaction tensor \bar{J}_{int} can be reduced to a scalar J_{int} , providing a direct visualization of its influence on the energy levels and the resulting ESR spectra. Figure S4 represents three possible cases for J_{int} being zero, positive, and negative with its magnitude smaller than the Zeeman energy, as experimentally realized in this work.

In Fig. 1c of the main text, the stronger signal appears at a higher frequency, which indicates the FM coupling in the given experimental condition, now schematized in Fig. S4b,e. As summarized in Fig. 1d, the coupling strength and polarity changes depending on the magnetic field direction due to the dominant contribution of the dipole interaction in the Ti-Er dimer with 0.928 nm separation. In the rest of figures in the main text, the Ti-Er dimer with 0.72 nm separation shows exceedingly higher contribution of antiferromagnetic exchange interactions than the dipole coupling, which shows the higher peaks at lower frequencies corresponding to the schematic in Fig. S4c,f.

Figure S4 | Influence of the Er-Ti interaction on the energy levels and ESR spectra. a,b,c, Four-level scheme of Er-Ti dimers with $J_{int} = 0$, $J_{int} < 0$ (ferromagnetic) and $J_{int} > 0$ (antiferromagnetic) respectively. d,e,f, Schematics of the resulting ESR spectra on Ti in its resonance frequency range. When no interaction is present (a) only one peak is detectable (d). A ferromagnetic interaction ($J_{int} < 0$) shifts the antiparallel levels to higher energies (b), resulting in two distinguishable peaks with the higher intensity one (f_1^{Ti}) at a higher frequency (e). An antiferromagnetic interaction ($J_{int} > 0$) shifts the antiparallel levels to lower energies (c), resulting in two peaks with the higher intensity one (f_1^{Ti}) at a lower frequency (f).

To further clarify this point, we added a sentence at page 3, which reads "When isolated, a nuclear spin-free Ti atom presents a single ESR signal under an external magnetic field (see Fig. S3a). The ESR peak of

Ti splits when coupled to an Er atom (Supplementary Section 4).” We also revised the caption of Fig. 1c, which now reads “For the latter, the relative peak intensity indicates a ferromagnetic interaction.” In addition, all over the text and SI we aligned the sign of J_{int} to match the previous literature, with positive/negative sign indicating antiferromagnetic/ferromagnetic coupling, respectively.

7. In Figure 3c: How do I have to picture an upward relaxation mechanism (Gamma_2)? Or are the levels drawn there no energy levels?

Reply: To clarify the reviewer’s point and explain this upward relaxation, we included a new section in the Supplementary Section 9 (rate equation model). Here we introduce a new schematic (Fig. S9) representing the four-level scheme and the respective populations for each level in 3 different cases: at thermal equilibrium (Fig. S9a), driving the system into resonance with f_3^{Er} excluding (Fig. S9b) and including relaxations (Fig. S9c). By driving f_3^{Er} at saturation, we can equalize the populations of the $|\downarrow\downarrow\rangle$ and $|\downarrow\uparrow\rangle$ state, which drives the system far from the thermal equilibrium state (Fig. S9b). Given that the spin relaxation of Er is much slower than the one of Ti, the system tends to relax through the latter (dotted purple arrows in Fig. S9b). Consequently the relaxation path starts from transferring the excess of population from $|\uparrow\downarrow\rangle$ to $|\downarrow\downarrow\rangle$, which is then transferred to $|\downarrow\uparrow\rangle$ by the excitation f_3^{Er} , and finally towards $|\downarrow\uparrow\rangle$ by the Ti upward relaxation mechanism, with the required energy transferred from the environment.

Figure S9 | Populations of the Er-Ti dimer calculated with the rate equation model at different conditions. **a**, Four-level scheme and respective populations for each level at thermal equilibrium. **b**, Driving the system into resonance at f_3^{Er} (shown as solid red arrow) and prior to activating the relaxation terms described in the model. The ESR driving leads to an equalization of n_{00} and n_{01} , which makes the Ti populations far from the Boltzmann distribution. Dashed arrows represent the relaxation path realized by the Ti relaxation mechanism, where part of the population of n_{10} is transferred to n_{00} through downward relaxation mechanism. The rf excitation re-equilibrate the excess of population towards n_{01} , which is further transferred to n_{11} via upward Ti relaxation mechanisms. **c**, Resulting populations after the inclusion of the relaxation terms, with the Er relaxation shown as dashed yellow arrows and the Ti relaxation as pink solid arrows.

We additionally revised the manuscript at page 9, which now reads “These relaxations happen due to an exchange of energy with the environment which tends to relax the populations towards the thermal equilibrium. An exchange of energy to (from) the environment leads to a transition to a lower (higher) energy level.”

8. Line 167: What are the significant digits and error bars of both of these g-factors? How were they determined? And as for the latter one, wasn’t it reported to be anisotropic?

Reply: The reviewer is correct. The g-factor for Ti is known to be anisotropic. Using the anisotropic g-tensor given in ref. [29], we calculated the g-factor projected along the applied magnetic field. Since the g-tensor in ref. [29] contains experimental errors, we gave the error values considering the error propagation. In contrast, the $g_{Er} = 1.2$ is the Er Landé g-factor calculated from its atomic quantum numbers, which gives no error bar. To clarify this, we revised the manuscript by mentioning the projection direction of g-value, error bar for Ti, and how we obtained the Er g-factor (page 4 and 7).

9. When reading lines 149/150 and 200, it is not fully clear to me how the peaks/dips behave for the ESR signal of the Er. Are the Ti and Er peaks always opposite (peak vs dip)? Or is the Er signal tip dependent and can be both, either a peak or a dip?

Reply: Both ESR signals are influenced by the STM tip, but in a different way. In our experiment, we observed only positive ESR signals for Ti. However, we observed that the polarity of Er ESR signals differs depending on the STM tip. For the Er, the ESR signal is detected through the Ti and, thus, we have to consider the spin relaxation processes of both Ti and Er and the spin pumping of Ti due to the spin-polarized electrons. In general, the ESR signal is obtained by modulating the RF signals and detecting the change in conductance. While there is no change in the junction conductance at the off-resonance frequency, the change occurs once the system is on-resonance. This can be expressed as given in Eq. S5:

$$\Delta I = C \cdot \{[(n_{10} + n_{11}) - (n_{00} + n_{01})] - [(n_{10} + n_{11}) - (n_{00} + n_{01})]_{\text{undriven}}\} \quad (\text{S5})$$

In our experiment, the STM tip is located over the Ti atom in the Ti-Er pair. For the Ti ESR signal, the polarity is determined by the relative spin polarization between Ti and the magnetic tip. The latter depends on the tip while the former stays almost the same at given experiment conditions. At tunnel currents with several tens of pA, this spin pumping does not play the major role for Ti ESR, since this process is much weaker and slower than the ESR-driving. However, for the indirect sensing, this spin pumping occurs in a similar time scale to the spin relaxation process and thus plays an important role in determining the ESR signal intensity and polarity.

To clarify this point, we specified in the main text that "...while the Ti transitions always yield positive peaks $f_{1,2}^{Ti}$, Er signals differ depending on specific tip conditions, i.e., different tips show positive or negative sign for $f_{3,4}^{Er}$." In addition, we modified Supplementary Section 9 to include Eq. S5 and the related discussion.

10. Line 219: Should the f4Ti not be f4Er?

Reply: The reviewer is correct. We appreciate the reviewer's careful reading. We corrected it accordingly.

Reviewer #3 (Remarks to the Author):

The authors present in this work an experimental approach to drive electron-spin resonance (ESR) on Er atoms (lanthanide atoms with 4f electrons) and find relatively long spin relaxation times, as compared with atoms of transition elements. The approach is based on the coupling of Er atoms with Ti atoms, which act as detectors.

The main conclusion of this work is of great importance for the ESR community and widens the possibilities to use single atoms to do electrically driven quantum coherent

control of single spins on surfaces. The work is scientifically sound, the manuscript is well written, and the main conclusions are clearly stated. So, in this sense, I am certainly inclined to recommend the publication of this manuscript provided several issues are clarified or better described in a revised version. Below, I proceed to list those issues.

1) For didactic reasons, and as a reference, it would have been good to add in the main manuscript ESR spectra of isolated Ti atoms. The authors could at least mention in a revised version that those spectra are reported in Fig. S3 in the SI. In general, I think that the authors could do a better job in the manuscript specifying where to find the information in the SI.

Reply: We are grateful to the Reviewer for supporting the publication of our paper. We agree that the manuscript can be improved by better referencing the SI content in the main text. Since the spectrum given in Fig. 1c is quite comparable to the single Ti ESR spectrum, we preferred not to add it to the main text. Instead, in the revised manuscript at page 6 we provide a more precise referencing to Fig. S3 as follows: "However, despite using a tip showing ESR signal on an isolated Ti atom (Fig. S3a), we observed no ESR when positioning the tip over an Er atom..." In addition, we further refer Fig. S5 in the main text (page 6) with the following sentence: "These transitions are not observed for all Er atoms, possibly due to the presence of isotopes with large nuclear spins for which the intensity of the ESR signal is spread over multiple peaks and is below the sensitivity of our measurements (Fig. S5)."

2) As the authors explained in the manuscript, no direct ESR signal was possible to acquire when they placed the SP-STM tip on top of the Er atoms. They also speculate about the reason for that. Maybe they can elaborate a bit more using the information provided in Fig. S1, namely it would be interesting to know if the differential conductance spectra in the absence of microwaves already hint at the absence of an ESR signal when the microwaves are applied.

Reply: The reviewer is correct. However, the absence of spectral features is not directly related to the absence of the ESR signal from Er. For example, the Ho on MgO shows no features in dI/dV spectra, but the spin switching of 4f electrons of Ho can be detected in the tunnel current. In fact, the absence of spectral features indicates no unpaired electrons in 6s/5d orbitals and therefore a weak coupling between the tunneling electrons and the 4f orbitals. To link the featureless spectrum of Er and follow the hint suggested by the reviewer, we revised the manuscript (page 6): "The weak polarization of the outer shell is reflected in the absence of spin excitations in the dI/dV spectra, as reported in Fig. S1d,e,h. These factors ..."

3) From the text in the manuscript, it is a little bit unclear what the authors mean by a term originating from the spin-polarized current in the context of the rate equation model summarized in Fig. 3c. I know that they elaborate on this in section 7 of the SI, but readers would benefit from a more detailed/didactic explanation of the meaning of this spin pumping.

Reply: As the reviewer suggested, we included a more detailed description about the spin pumping at page 9 of the revised manuscript, which reads: "In addition, to account for the tip-dependent sign and intensity of Er ESR signals, we included a spin pumping term originating from the spin-polarized tunnel current (Fig. 3c for a negatively polarized tip). In inelastic scattering events, the exchange of angular momenta occurs while retaining the total angular momentum of the system. That is, the spin-polarized tunneling electrons lead to scattering events with preferential polarization, as depicted in the inset of Fig. 3a and

Fig. 3c. Thus, the tunneling electrons can shift the Ti spin occupation altering the population balance with respect to the thermal equilibrium (see Supplementary Section 9)."

4) Related to the previous issue, in the discussion of the rate equation model (section 7 of the SI), it is a bit unclear to me how the solution of the rate equation is finally converted into the ESR signal (ΔI in Fig. S7). Could the authors elaborate more in the SI about this issue?

Reply: As the reviewer suggested, we added an additional section in the Supplementary Section 9 explaining how we calculated the ESR intensity from the rate equation model. The newly added equations (Eq. S5 and S6) show how the populations of each level are converted to the ESR signals.

5) The authors measure the relaxation time in the Er atoms with a sophisticated protocol that I must admit I have not quite understood. Could the authors make this discussion at the beginning of section "Relaxation Time Measurement ..." somehow more accessible to non-specialists?

Reply: Following the reviewer's suggestion, we added detailed descriptions at the beginning of the section at page 11 which reads: "As previously discussed about the Er-Ti dimer with 0.928 nm separation, the relative peak intensity of the Ti peaks f_1^{Ti} and f_2^{Ti} reflects the time averaged population of the Er states (Fig. 4a), i.e. changes in the time-averaged spin states of Er will induce the change of the relative peak intensity of f_1^{Ti} and f_2^{Ti} . Thus, by monitoring the Ti ESR signals, we can observe the evolution of the Er spin state. In this way, we characterize the characteristic relaxation time T_1^{Er} by exciting the Er into an out-of-equilibrium state and monitoring its relaxation towards the thermal state." In addition, in the following paragraph, we added one sentence, which reads: "With this scheme, it is possible to drive Er transitions and sense the change in population through the Ti ones."

6) Related to the previous issue, the authors report the Er relaxation time as a single number. Should I understand that this is the intrinsic relaxation time of Er on this surface? How can one avoid the influence of the Ti atom?

Reply: In general, it is difficult to access the intrinsic relaxation time of a certain material system since there are always interactions with the environment: a higher isolation of the system leads to a longer relaxation time. In our experiment, we did isolate the Er from the direct impact of the tunneling electrons, which is one of the main relaxation sources in STM experiments. However, the Er is strongly coupled to the Ti atom (about 2.7 GHz), which exposes the Er states to the Ti spin noise, as generated by the rapidly fluctuating spin under the influence of tunneling electrons. Thus, as the reviewer pointed out, the measurement result is not the intrinsic relaxation time of Er on this surface. To avoid this, one should probably build different atomic structures to reduce the spin noise. To include the reviewer's point, we revised the manuscript at page 11, which now reads: "The T_1^{Er} observed through Ti likely differs from the intrinsic relaxation time of Er on this surface due to its strong interaction with the Ti atom. Nevertheless, the large T_1^{Er} ..."

In addition, in the conclusion at page 13 we added a perspective sentence as follows: "We expect that, by employing a similar approach to different atomic structures, we can reduce the influence of the spin fluctuations of the atom used for the detection..."

Reviewers' Comments:

Reviewer #2:

Remarks to the Author:

I thank the authors for addressing all my previous concerns I had with the manuscript. The answers have improved the manuscript and help clarifying questions I had, and a potential reader might have as well. Especially the additional explanation of models, what ESR peaks to expect and the softening of some claims is appreciated.

I only have one small remark left, that I would ask the authors to adjust.

3. I am not fully in line with the added sentence in the methods about the Ti atoms. Even though the authors are using the bridge site Ti species, I find the sentence now too unprecise. I much prefer what has been written in line 71, namely that the spin magnitude is $\hbar/2$ and that there is a (weak) g-factor anisotropy. The added sentence in the methods now suggests the Ti being a pure spin $1/2$ atom, to which I strongly disagree. Especially since the added text does not specify the adsorption site. I'd suggest rephrasing this to be similarly as has been written in line 71 of the manuscript.

Reviewer #3:

Remarks to the Author:

The authors have certainly made an effort to address all the issues that I raised in my first report and, in particular, I am satisfied with the answers and the changes introduced accordingly both in the manuscript and in the supplementary information. So, in this sense, I reiterate here my support for the publication of the this manuscript, which is going to be welcome by the ESR-STM community.

Reply to Reviewers

Reviewer #2 (Remarks to the Author):

I thank the authors for addressing all my previous concerns I had with the manuscript. The answers have improved the manuscript and help clarifying questions I had, and a potential reader might have as well. Especially the additional explanation of models, what ESR peaks to expect and the softening of some claims is appreciated.

I only have one small remark left, that I would ask the authors to adjust.

3. I am not fully in line with the added sentence in the methods about the Ti atoms. Even though the authors are using the bridge site Ti species, I find the sentence now too unprecise. I much prefer what has been written in line 71, namely that the spin magnitude is $\hbar/2$ and that there is a (weak) g-factor anisotropy. The added sentence in the methods now suggests the Ti being a pure spin 1/2 atom, to which I strongly disagree. Especially since the added text does not specify the adsorption site. I'd suggest rephrasing this to be similarly as has been written in line 71 of the manuscript.

Reply: We thank the reviewer for his/her contribution in improving the manuscript. We modified the sentence in the methods to make it more precise. The sentence now reads: "As described in previous works, the Ti atoms on 2 ML MgO/Ag(100) show a spin magnitude of $\hbar/2$ with g-factor anisotropy²⁹, a behavior previously attributed to hydrogenation³¹."

Reviewer #3 (Remarks to the Author):

The authors have certainly made an effort to address all the issues that I raised in my first report and, in particular, I am satisfied with the answers and the changes introduced accordingly both in the manuscript and in the supplementary information. So, in this sense, I reiterate here my support for the publication of the this manuscript, which is going to be welcome by the ESR-STM community.

Reply: We thank the reviewer for her/his valuable comments and for supporting the publication.